# Comparing CNNs and Random Forests for Landsat Image Segmentation Trained on a Large Proxy Land Cover Dataset

Tony Boston *[ID], Albert Van Dijk [ID], Pablo Rozas Larraondo [ID] and Richard Thackway [ID]

Fenner School of Environment and Society, Australian National University, Canberra, ACT 2601, Australia; albert.vandijk@anu.edu.au (A.V.D.); pablo.larraondo@anu.edu.au (P.R.L.); richard.thackway@anu.edu.au (R.T.)

* Correspondence: tony.boston@anu.edu.au

**Abstract:** Land cover mapping from satellite images has progressed from visual and statistical approaches to Random Forests (RFs) and, more recently, advanced image recognition techniques such as convolutional neural networks (CNNs). CNNs have a conceptual benefit over RFs in recognising spatial feature context, but potentially at the cost of reduced spatial detail. We tested the use of CNNs for improved land cover mapping based on Landsat data, compared with RFs, for a study area of approximately 500 km × 500 km in southeastern Australia. Landsat 8 geomedian composite surface reflectances were available for 2018. Label data were a simple nine-member land cover classification derived from reference land use mapping (Catchment Scale Land Use of Australia—CLUM), and further enhanced by using custom forest extent mapping (Forests of Australia). Experiments were undertaken testing U-Net CNN for segmentation of Landsat 8 geomedian imagery to determine the optimal combination of input Landsat 8 bands. The results were compared with those from a simple autoencoder as well as an RF model. Segmentation test results for the best performing U-Net CNN models produced an overall accuracy of 79% and weighted-mean F1 score of 77% (9 band input) or 76% (6 band input) for a simple nine-member land cover classification, compared with 73% and 68% (6 band input), respectively, for the best RF model. We conclude that U-Net CNN models can generate annual land cover maps with good accuracy from proxy training data, and can also be used for quality control or improvement of existing land cover products.

**Keywords:** land cover; Landsat data; segmentation; convolutional neural networks; random forests; deep learning; transfer learning

## 1. Introduction

Better understanding and management of the Earth's land area is increasingly important in the face of human-induced global change, urbanisation, growing population and environmental pressures [1]. Land cover, the 'observed (bio)physical cover on the Earth's surface' [2], is an essential climate variable [3] and essential biodiversity variable [4], critical to improved understanding of the nature and extent of land resources, and their dynamics and transformations. In Australia, there are relevant national mapping initiatives. The Catchment Scale Land Use of Australia (CLUM) [5] collates land use mapping data. The National Carbon Accounting System (NCAS) Land Cover Change Mapping [6,7] produces binary forest/non-forest mapping. The Forests of Australia [8] mapping produces forest type and purpose mapping. Finally, the National Vegetation Information System (NVIS) [9,10] provides data on vegetation communities. This mapping is at resolutions of between 50–100 m, often partly based on interpretation of Landsat imagery and other contributions from national, state and territory government agencies and external sources. Their ongoing generation requires considerable manual effort. Update cycles are generally infrequent and importantly, reflect changes in the landscape but also in mapping methods. Previous comparisons have shown inconsistencies and errors between these data products [11].

Van Dijk and Summers [12] found that incomplete or poor-quality vegetation change mapping is one of the greatest obstacles to monitoring environmental conditions across Australia. Monitoring land cover change is critical for several environmental policies [13] and lacking or inconsistent data are a major issue. For example, inconsistent numbers on forest extent and land clearing in Australia from different sources have raised questions about the validity of carbon emissions reporting [14,15]. Transparent and accurate techniques are needed to detect and attribute vegetation change.

Land cover mapping from satellite images has progressed from visual and simple statistical approaches to artificial neural networks (NNs) [16], Random Forests (RFs) and, more recently, advanced image recognition techniques such as convolutional neural networks (CNNs). Random Forests [17] are a leading method for generating land cover classifications [18,19]. They achieve similar or superior accuracy than more traditional classifiers with better computational performance [20,21] for both pixel-based classification and object-based image analysis (OBIA) [22]. For these reasons, Random Forest models are often used as a benchmark against which to compare results generated using deep learning, as undertaken in this study.

Since 2012, deep learning CNNs have increasingly been applied to spatial reasoning, such as image classification or semantic segmentation [23–25]. CNNs discover the most discriminative features in an image and exploit the surrounding context with weights of the neural network learnt through an iterative stochastic gradient descent algorithm over a number of epochs. CNNs typically consist of a sequence of three layers: convolution, nonlinearity and pooling, repeated several or many times, followed by a final fully connected layer to determine the class with the highest probability. CNNs have the advantages of being non-parametric, generally quick and easy to configure, and able to handle large volumes of data with low risk of overfitting [26].

U-Net is a CNN implementation originally developed for biomedical image interpretation [27] but it has also demonstrated cutting-edge results in satellite image segmentation. A review of the literature, using key words in Scopus searches, showed that U-Net is currently the most-used deep learning architecture for segmentation of remote sensing images, and for this reason it was chosen for model-building in this study.

CNNs have achieved good results for many image analysis tasks such as land use and land cover (LULC) classification, scene classification, as well as object detection, and the median accuracy of the LULC classifications using deep learning methods is higher than that of other classifiers such as Random Forest (RF) decision trees or support vector machines (SVM) reflecting their ability to exploit, through convolutional layers, spatial/textural features as well as spectral information [28,29]. CNNs' ability to extract information from spatial patterns suggests, and has commonly led to, their application mainly to high resolution imagery [28,30,31]. In this study, we wanted instead to test their application to Landsat imagery, potentially supporting future use with this global long time series open access data to measure land cover change. We compared RFs and NNs (whether using a simple autoencoder or the U-Net CNN) for the production of land cover maps using Landsat imagery, and which combination of inputs, models and configurations produces the best results.

A major limitation to any deep learning land cover classification method using medium resolution satellite imagery such as Landsat is the lack of availability of benchmark training datasets, and limited availability of high-quality training data can reduce the effectiveness of data-hungry machine learning methods such as CNNs [28]. To address this, we combined two existing mapping data products to produce a 'proxy' dataset of land cover. This has the benefit of producing a very large training dataset, but has the potential disadvantage of introducing errors associated with the original mapping as well as its translation from proxy to actual land cover.

Our initial hypotheses were:

1. U-Net CNNs can produce land cover maps of comparable or better accuracy than RFs; and

2. An imperfect proxy variable can be used to train a well-performing machine-learning classifier.

## 2. Material and Methods

### 2.1. Input Data Preparation

The input data used were based on pixel-based image compositing [32,33]. Such image composites on a national basis support automation of a wide range of image classification methods and solve the problem of selection and access to cloud-free imagery [34,35]. Pixel-based compositing methods are increasingly being recognised as a valuable remote sensing technique to leverage the large volumes of available data by creation of representative images for annual or seasonal time periods [33] as input to model development. Annual geomedian data [36] contain pixel composites of Landsat images using a high dimensional statistic called the 'geometric median', providing a method of combining images that produces a representative cloud-free annual composite image maintaining the 'spectral relationship between bands, reduced spatial noise, and consistency across scene boundaries' [37]. Annual Landsat 8 geomedians were available from Digital Earth Australia (DEA) for 2018 at 25 m resolution. Twenty-five Landsat tiles in GDA94 projection from NSW, ACT and northern Victoria (each approximately 100 km × 100 km) were selected for the experiment, with 24 used for model training and validation (RGB image in Figure 1) and one (tile +14, −40 annotated with an 'X' in Figure 1) used for model testing.

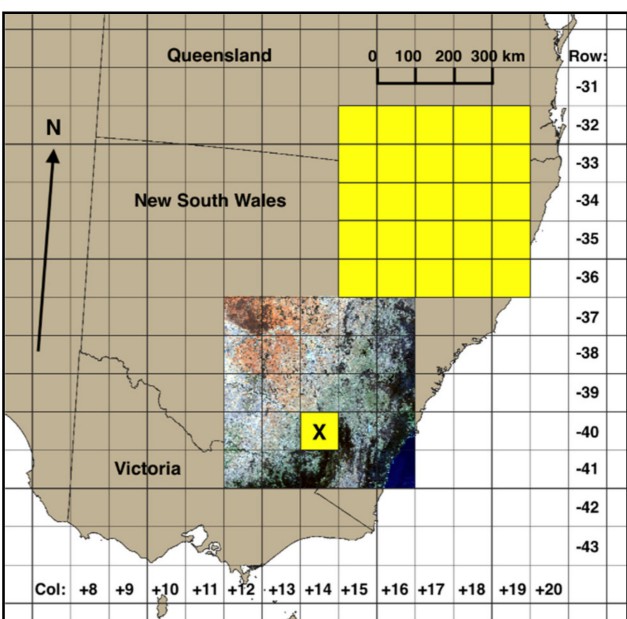

**Figure 1.** Study area (2018 Landsat 8 geomedian true colour RGB), test tile +14, −40 (annotated with 'X') and northern test area in yellow. Projection: GDA94/Australian Albers (EPSG: 3577).

The DEA data cube also provided the Triple Median Absolute Deviation (TMAD), describing the median of absolute deviations from the geomedian, which provides information on annual temporal variance through three calculations of distance: Euclidean distance (edev), spectral distance (sdev) and Bray–Curtis dissimilarity (bcdev) [38].

For the segmentation experiment, 2018 Landsat 8 geomedian data for the study area were prepared as a nine band (six Landsat bands plus three TMADs bands) GeoTIFF image. All bands were encoded as uint16 with any 'No data' pixels in the images converted to zero values. The base GeoTIFF was broken into 128 × 128 tiles and these were converted into NumPy array format [39] for use in modelling.

## 2.2. Label Data Preparation

Label (reference) data were created using the Australian Collaborative Land Use and Management Program's (ACLUMP) 2018 Catchment Scale Land Use of Australia (CLUM) 50-m resolution gridded data [40]. ACLUMP provides nationally consistent land use mapping compiled by ABARES from state/territory contributions at catchment scale [5]. Although nominally for 2018, the data represent state/territory mapping captured over a range of years: NSW in 2017, ACT in 2012 and Victoria in 2016 or 2017. CLUM uses the hierarchical ALUMv8 classification system to describe land use classes [41]. We converted the CLUM land use data into land cover through the process described below. Although by no means an error-free conversion, the resulting land cover label data were expected to be sufficiently accurate to train machine learning models.

The 6 primary and 32 secondary classification levels in ALUMv2 relate to principal land use, while the 159 tertiary classes may include 'additional information on commodity groups, specific commodities, land management practices or vegetation information' [41]. We translated the 191 secondary and tertiary ALUMv8 classes to a simple, nine-member land cover classification—Bare, Built-up, Crop, Forest, Grassland, Horticulture, Plantation, Unknown and Water—using a class correspondence table (Table S1). The Unknown class included land uses, e.g., nature conservation, for which no clear relationship with land cover is expected.

Subsequent work focused on reducing the presence of the Unknown class. We identified forest areas based on Forests of Australia [8] to improve land cover mapping in ambiguous classes such as nature conservation, managed resource protection, other minimal use, grazing native vegetation and residential and farm infrastructure. We manually assigned land cover classes to surface water supply areas which included water storages but sometimes also surrounding areas of vegetation. Finally, we visually compared all larger areas mapped as Unknown to Landsat imagery to determine any that could be unambiguously assigned to a land cover class based on their spectral and visual characteristics. The resulting 50 m resolution label data were resampled to 25 m to match the Landsat image resolution (Figure 2).

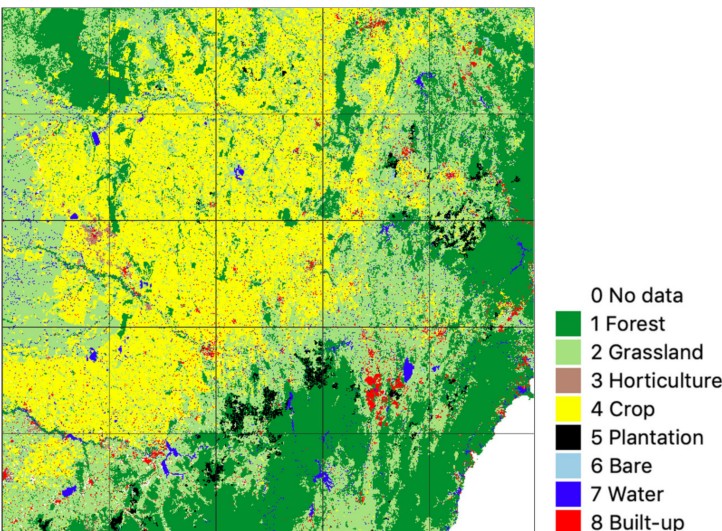

**Figure 2.** Label data (circa 2018)—simplified land cover map for study area (~500 km × 500 km) derived from CLUM [40]. Projection: GDA94/Australian Albers (EPSG: 3577).

Tile +14, −40 (indicated with an 'X' in Figure 1) was excluded from the experiment as test data, to calculate accuracy and other metrics for alternative models. The pixel distribution of land cover classes for the whole study area and the test tile are listed in the Supplementary Materials (Table S2). We considered the test tile to be reasonably

representative: the largest differences compared with the overall study area were that it had more plantation (11.4% vs. 1.42%) and less cropland (15.8% vs. 29.7%).

### 2.3. Modelling Approach

Large labelled remote sensing training datasets were rare when CNNs became popular in 2012, with early applications often using models pre-trained on computer vision datasets such as ImageNet [42]. The assumption behind this 'transfer learning' approach [43] is that the model has been trained on a sufficiently large dataset to serve as an effective generic model for visual recognition. Transfer learning provides a method to speed up training and boost accuracy, especially on remote sensing datasets which are often imbalanced, with uneven distributions of classes in terms of their frequency, extent and spatial scale, as well as heterogeneity, reflecting greater intra-class than inter-class variation [44]. Use of transfer learning has shown surprisingly good results in remote sensing image classification, even when using models pre-trained in a different domain [45,46].

Segmentation models aim to find a function that effectively maps an input (here, Landsat 8 derived data) to an output (here, the labelled land cover data derived from CLUM) by maximising accuracy and minimising loss. Land cover maps were created using the two different approaches, i.e., RFs and NNs.

Experiments were carried out with the following algorithms:

1.  Autoencoder: a simple six-layer neural network that learns a representation of reduced dimensionality then decodes it to the original image dimensions. The last layer of the neural network predicts a class label of the pixel of interest (Figure S1);

2.  U-Net: a CNN encoder–decoder model where symmetric connections between the convolution and deconvolution layers are used to capture spatial context and build a final segmented image of the same resolution as the input. The depth of the base U-Net model was five, with dimension reductions of $128 \times 128$, $64 \times 64$, $32 \times 32$, $16 \times 16$, $8 \times 8$ (Figure 3). The CNNs were coded using the segmentation modelling software developed by [47]; and

3.  Random Forests (RF): an algorithm for classification and regression in which decision tree classifiers are fitted on various subsets of the dataset based on an attribute test. Predictions for each pixel were output by the RF classifier trees based on the class label that received majority support.

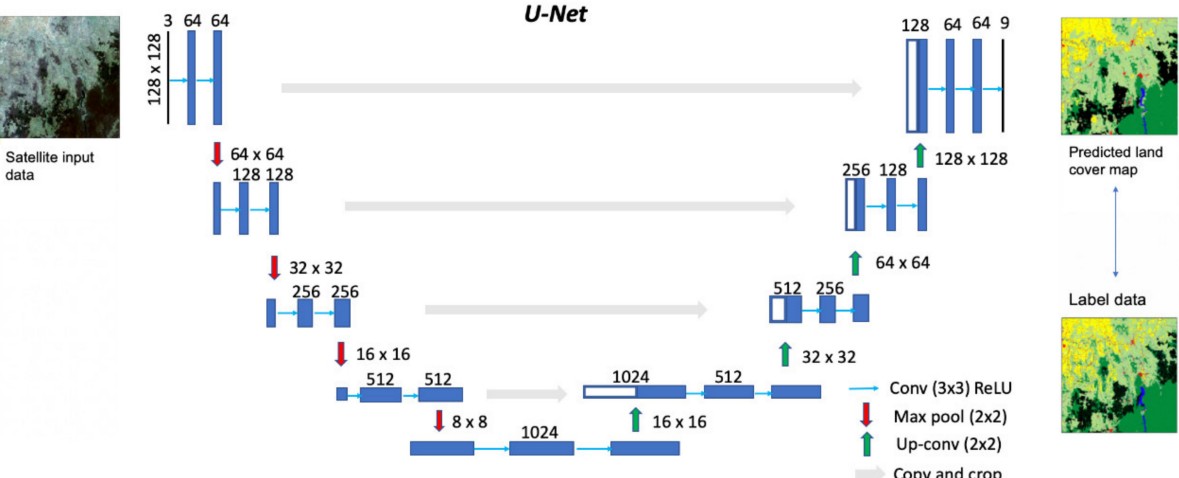

**Figure 3.** Base U-Net architecture after Ronneberger et al. (2015, Figure 1) [27].

The U-Net CNN models were evaluated for six combinations of Landsat 8 inputs in an attempt to better understand which bands have the most information for segmentation, and how overall model performance relates to geomedian inputs and combinations:

1.  Only one of the spectral bands, i.e., red (R), green (G), blue (B), near-infrared (N), or one of the two shortwave infrared bands (S1 and S2);
2.  Only one of the TMADs bands: Euclidean distance (edev), Spectral distance (sdev) and Bray–Curtis dissimilarity (bcdev);
3.  Selected combinations of three bands: RGB, NS1S2 and TMADs;
4.  The three visible bands plus near-infrared (RGBN);
5.  All six bands; and
6.  All six bands and three TMADs.

For the U-Net CNNs, transfer learning [43] was employed using various ResNet models [48] as the encoder portion of the U-Net model, initialised with weights obtained from training using the ImageNet ILSVRC-2012-CLS dataset [42]. Several remote sensing studies have used ImageNet-trained ResNet encoder weights with the U-Net models for segmentation to produce land cover maps, e.g., [49–51]. We compared the results from ResNet encoder versions with varying numbers of neural network layers (ResNet18, ResNet34, ResNet50, ResNet101, ResNet152).

The U-Net CNN model required a minimum of three input channels. For model builds using a single band, the band was, therefore, duplicated three times. These three channels were initialised with pre-trained weights from ImageNet and any additional channels for 4, 6 or 9 band input needed to be learnt from scratch. All neural network models were trained using the Adam optimiser [52] with a learning rate of 0.0001 for 80 epochs with a batch size of 8. Dice and focal loss functions were used to manage imbalanced class distributions following [53]:

$$\text{Total\_loss} = \text{Dice\_loss} + (1 \times \text{Focal\_loss})$$

Inputs to model building were $128 \times 128$ pixel tiles of Landsat geomedian and label data. Of the 24,336 non-overlapping tiles for the study area, 694 were over the ocean and discarded. The remainder were split into train, validation and test sets as described in Table 1.

**Table 1.** Dataset splits used for model development. Test set from tile +14, −40 (Figure 1).

| Dataset Splits | Count | Percent |
|:---:|:---:|:---:|
| Train | 19,086 | 81% |
| Validation | 3532 | 15% |
| Test | 1024 | 4% |
| TOTAL | 23,642 | 100% |

Data augmentation increases dataset size and can reduce overfitting, and was undertaken using the fast augmentation library Albumentations [54] applying various transforms, including flipping, mirroring, affine transforms, perspective transforms, image blurring and sharpening, gaussian noise addition and random cropping. Models were built for ten epochs at a time. Those with the highest validation accuracy were saved and run for the test tile (+14, −40; Figure 1) to evaluate performance of the predicted land cover classification with label data through visual inspection. The keras-based callback ReduceLROnPlateau [55] was used to reduce learning rate after ten epochs if validation accuracy failed to improve.

Random Forest modelling was undertaken using the Python scikit-learn package [56]. Several variations of RF parameters were tested, including the number of trees/estimators (50, 100, 200, 300) and maximum depth (10, 20, 30, 40, 50). A random selection of 0.2% (800,000) pixels were used as input to the RF classifier to remain within memory limitations, selected using three different sampling strategies: random sampling of pixels (RSP), stratified random sampling proportional to area occupied by each class (SRS-Prop), and stratified random sampling with equal samples for all classes (SRS-Eq).

The accuracy and performance of land cover classifications were assessed using baseline statistical measures [57] able to be derived from a confusion matrix. The main measures used were the overall accuracy (OA), and per class user's accuracy (UA), producer's accuracy (PA), and F1 scores. The OA is the fraction of correctly classified pixels:

$$\text{Overall accuracy (OA)} = \frac{\text{Correct pixels}}{\text{All pixels}} = \frac{\text{TP} + \text{TN}}{\text{TP} + \text{TN} + \text{FP} + \text{FN}}$$

F1 is the harmonic mean of precision (UA) and recall (PA) defined as:

$$\text{F1} = 2 \times \frac{\text{precision} \times \text{recall}}{\text{precision} + \text{recall}} = 2 \times \frac{\text{UA} \times \text{PA}}{\text{UA} + \text{PA}}$$

where $\text{UA} = \frac{\text{TP}}{\text{TP}+\text{FP}}$ and $\text{PA} = \frac{\text{TP}}{\text{TP}+\text{FN}}$, and TP are true positives, TN: true negatives, FP: false positives, and FN: false negatives.

There are several potential sources of error in the label data, including errors in the original land use mapping, errors during compilation of the CLUM data, and errors relating to the translation of CLUM land use classes to the simplified land cover classification applied here. In order to quantify these errors, 100 pixels for each of the eight land cover classes were randomly selected for the study area (Figure 4) and examined manually to determine if they were the correct class (Section 3.1). Validation was carried out through visual inspection of 2018 Landsat geomedian data or, if not clear in this imagery, publicly available higher resolution sources such as Google Earth or SPOT 5 [58]. Confusion matrix statistics were generated to estimate the overall accuracy and other metrics for label data and the U-Net CNN model based on these 800 sample points.

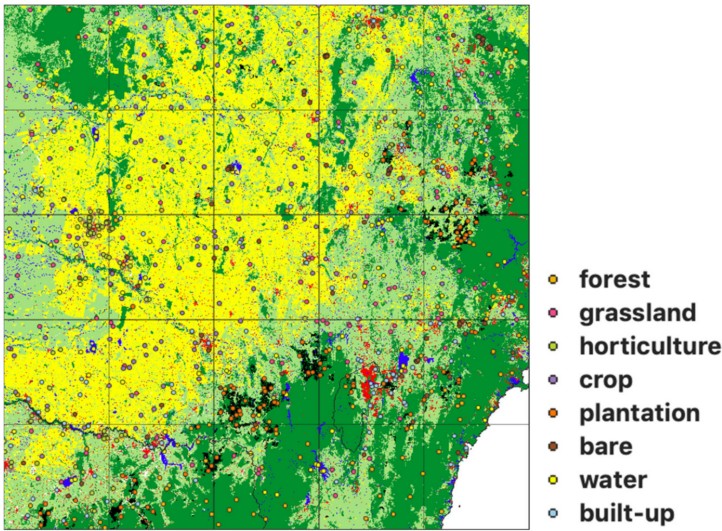

**Figure 4.** The 800 sample points of the study area (~500 km × 500 km) to evaluate label data and U-Net CNN model. See Figure 2 for colour legend of base map.

Finally, as a test of spatial transferability, the U-Net CNN model was used to generate land cover maps (individually, due to memory limitations) for each of the 25 tiles (yellow area in Figure 1) northeast of the study area and stitched together (Section 4.2).

## 3. Results

The overall accuracy (OA) and per-class F1 scores corresponding to all segmentation experiments are listed in the Supplementary (Table S3). Overall accuracies ranged from 72–79%, with weighted-mean F1 scores of 59–77% for 22 model builds. The simple autoencoder model produced the lowest OA and F1 scores of 72% and 59%, respectively, with better results produced by the best RF (73% and 68%, respectively).

For the U-Net CNN, the highest validation accuracy was obtained after 12 to 79 epochs for different model builds. Typically, models failed to improve after reaching their optimum solution. U-Net with a ResNet50 encoder with ImageNet weights (Figure S2) produced the greatest OA in the six-band input experiment, when compared with other model configurations (Table S3). Generally, model performance improved as more bands were provided as input (Figure 5).

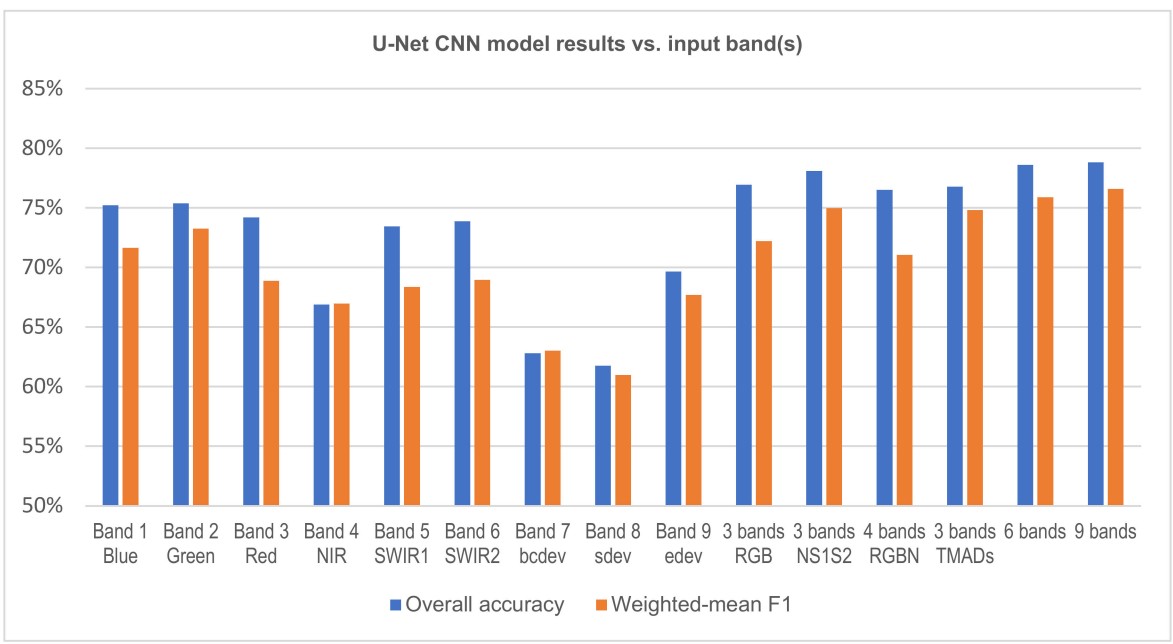

**Figure 5.** Impact of Landsat input bands on U-Net CNN model OA and F1 scores.

The greatest OA and F1 were obtained for a model with nine input bands (six Landsat 8 bands plus three TMADs bands) (bm_r1_urn50_9b_e34.h5 in Table S3 in the Supplementary Materials: OA = 79% and weighted-mean F1 = 77%). The next best result was obtained with six bands (bm_r2_urn50_6b_e36.h5: OA = 79% and F1 = 76%). Both used a U-Net CNN with ResNet50 encoder and ImageNet weights. We found that a generic U-Net model [27] with no transfer learning produced only marginally lower results (bm_r1_unet_6b_e79.h5: OA = 78% and F1 = 74%). It required more epochs to reach this accuracy (79 epochs compared with a mean of 31 epochs for the other U-Net CNN models), which may be explained by the fact that the model was built without pre-trained weights.

Among the models with a single band input, the green band produced the best scores (Supplementary Materials Table S3; OA = 75% and F1 = 73%). For the models with three band inputs, the combination of near- and shortwave infrared bands (NS1S2; OA = 78% and F1 = 75%) performed better than only visible bands (RGB; OA = 77%; F1 = 72%) or TMADs (OA = 77%; F1 = 75%), although differences were small. Adding near-infrared to RGB to create a four-band input did not improve results in this experiment (RGBN: OA = 77%; F1 = 71%).

Confusion matrices (Table S4) for test tile +14, −40 for the best performing U-Net CNN model (bm_r1_urn50_9b_e34.h5) showed the highest values of PA (probability that label land cover classes are correctly classified) and UA (how often predicted land cover classes are actually present on the ground) for Forest (PA = 80%; UA = 95%), Grassland (PA = 86%; UA = 75%), and Plantation (PA = 88%; UA = 76%). The worst performing classes were Horticulture (PA = 43%; UA = 20%) and Bare (PA = 42%; UA = 11%).

Although these results appear to not completely meet well-known benchmarks for land cover classifications such as 85% OA and 70% per-class accuracy (e.g., [59]), they were broadly in line with similar studies that use label data derived from land use or land cover mapping in combination with Landsat or Sentinel-2 inputs (e.g., [49,50,60–62]).

For the Random Forests modelling (Table S5), we used 800,000 pixels (ca. 0.2% of the total) using the same input six band Landsat 8 geomedian reflectances. Of the three sampling strategies, random sampling of pixels (RSP) produced the highest OA and F1 scores, with slightly better results than stratified random sampling proportional to area occupied by each class (SRS-Prop), which was, in turn, significantly better than the stratified random sampling with equal samples for all classes (SRS-Eq). The best results for RSP were obtained with 200 trees and the maximum depth of the tree of 30 (OA = 73.1% and F1 = 68%). The results from this model (ex4.1_rf_rsp_e200_d30.sav) were compared with those from the U-Net CNN models (Table S3). Model building using Random Forests is very efficient, able to be undertaken in a fraction of the time required for deep learning CNN models. Model builds of 80 epochs for the U-Net CNN took approximately eight hours on a single GPU machine, while model builds for Random Forest on a single CPU took approximately five minutes for 800,000 (0.2%) of study area pixels.

### 3.1. Accuracy of Label Data and CNN Model Output

Based on a random sample of 800 points, the accuracy of both label data and U-Net CNN model output (bm_r1_urn50_9b_e34.h5) were analysed to obtain overall and per class accuracies. The results were used to calculate per class user's accuracy (precision) and producer's accuracy (recall), F1 scores as well as overall accuracy, weighted-mean F1 and kappa coefficient ($\kappa$) [63].

The U-Net CNN model (OA = 83%) was ca. 4% more accurate than the label data (OA = 79%) (Tables 2 and 3). The CNN model also showed higher weighted-mean F1 score (82.3% vs. 74.7%) and kappa (80.2% vs. 76%). User's class accuracies for label data and U-Net CNN model (respectively) were calculated for Forest (93% vs. 87.8%), Grassland (80% vs. 78.7%), Horticulture (76% vs. 89.7%), Crop (92% vs. 74.1%), Plantation (92% vs. 93.2%), Bare (84% vs. 90.5%), Water (57% vs. 90.2%), and Built-up (58% vs. 73%).

In order to assess how the magnitude of label data errors impact CNN prediction accuracy, we determined the percentage of CNN model predictions that were correct for those sample points where the label data matched the visually assessed land cover class. It appears that the CNN model performed approximately the same or better for sample points where the label data were correct, and we concluded that the magnitude of label data errors impacted CNN prediction accuracy (Table 4). The weighted-mean class accuracy of the CNN model for sample points that were correctly labelled ($N$ = 632) was 93.4%, and the overall accuracy was 86.4%. These values were 85.4% and 83.0%, respectively, for all 800 sample points. For classes Forest (89.2% vs. 77.7% for all 100 pixels), Grassland (95.0% vs. 84.2%), Crop (97.8% vs. 94.8%) and Built-up (87.9% vs. 84.4%) there was clear improvement in CNN model class accuracy when the label data were correct. For the other classes (together representing less than 2% of pixels) the results were very similar. This result suggests that ground truth (label data) accuracy is important when developing CNN models, especially for the dominant classes. Another possible explanation is that regions where the label and model class are both correct are inherently more easily classified.

**Table 2.** Confusion matrix comparing visually identified class (rows) to label data class (columns). Rows indicate 'true' visually identified classes and columns indicate classes according to mapping. Matching class counts (diagonal), totals and primary metrics are in bold.

| Class Count | Label Class | | | | | | | | | Producer's Acc. | Omission | |
|---|---|---|---|---|---|---|---|---|---|---|---|---|
| **Vis. Id. Class** | **Forest** | **Grassland** | **Horticulture** | **Crop** | **Plantation** | **Bare** | **Water** | **Built-Up** | **Total** | **(Recall)** | **Errors** | **Total** |
| **Forest** | **93** | 8 | 5 | | 2 | 4 | 12 | 6 | **130** | 71.5% | 28.5% | 100.0% |
| **Grassland** | 5 | **80** | 9 | 5 | 5 | 9 | 25 | 33 | **171** | 46.8% | 53.2% | 100.0% |
| **Horticulture** | | 1 | **76** | | | | | | **77** | 98.7% | 1.3% | 100.0% |
| **Crop** | 1 | 10 | 6 | **92** | | 1 | 2 | 3 | **115** | 80.0% | 20.0% | 100.0% |
| **Plantation** | 1 | | 1 | | **92** | 2 | 1 | | **97** | 94.8% | 5.2% | 100.0% |
| **Bare** | | | | | 1 | **84** | 2 | | **87** | 96.6% | 3.4% | 100.0% |
| **Water** | | | | 2 | | | **57** | | **59** | 96.6% | 3.4% | 100.0% |
| **Built-up** | | 1 | 3 | 1 | | | 1 | **58** | **64** | 90.6% | 9.4% | 100.0% |
| **Total** | **100** | **100** | **100** | **100** | **100** | **100** | **100** | **100** | **800** | **79.0%** | **Overall Accuracy** | |
| **User's Acc. (Precision)** | 93.0% | 80.0% | 76.0% | 92.0% | 92.0% | 84.0% | 57.0% | 58.0% | | | | |
| **Commission Errors** | 7.0% | 20.0% | 24.0% | 8.0% | 8.0% | 16.0% | 43.0% | 42.0% | | | | |
| **Total** | 100.0% | 100.0% | 100.0% | 100.0% | 100.0% | 100.0% | 100.0% | 100.0% | | | **76.0%** | **Kappa** |
| **F1** | 80.9% | 59.0% | 85.9% | 85.6% | 93.4% | 89.8% | 71.7% | 70.7% | | | **74.7%** | **Weighted-Mean F1** |

**Table 3.** Confusion matrix comparing visually identified class (rows) to model prediction (columns). Matching class counts (diagonal), totals and primary metrics are in bold.

| Class Count | Model Prediction | | | | | | | | | Producer's Acc. (Recall) | Omission Errors | Total |
|---|---|---|---|---|---|---|---|---|---|---|---|---|
| Vis. Id. class | Forest | Grassland | Horticulture | Crop | Plantation | Bare | Water | Built-Up | Total | | | |
| **Forest** | **101** | 9 | 2 | 3 | 4 | 3 | 2 | 6 | **130** | 77.7% | 22.3% | 100.0% |
| **Grassland** | 4 | **144** | 3 | 11 | 2 | 1 | 1 | 5 | **171** | 84.2% | 15.8% | 100.0% |
| **Horticulture** | 1 | 10 | **52** | 12 | | | | 2 | **77** | 67.5% | 32.5% | 100.0% |
| **Crop** | | 6 | | **109** | | | | | **115** | 94.8% | 5.2% | 100.0% |
| **Plantation** | 5 | 6 | 1 | 1 | **82** | 1 | | 1 | **97** | 84.5% | 15.5% | 100.0% |
| **Bare** | | 2 | | 2 | | **76** | 1 | 6 | **87** | 87.4% | 12.6% | 100.0% |
| **Water** | 3 | 2 | | 5 | | 3 | **46** | | **59** | 78.0% | 22.0% | 100.0% |
| **Built-up** | 1 | 4 | | 4 | | | 1 | **54** | **64** | 84.4% | 15.6% | 100.0% |
| **Total** | **115** | **183** | **58** | **147** | **88** | **84** | **51** | **74** | **800** | **83.0%** | **Overall Accuracy** | |
| **User's Acc. (Precision)** | 87.8% | 78.7% | 89.7% | 74.1% | 93.2% | 90.5% | 90.2% | 73.0% | | | | |
| **Commission Errors** | 12.2% | 21.3% | 10.3% | 25.9% | 6.8% | 9.5% | 9.8% | 27.0% | | | | |
| **Total** | 100.0% | 100.0% | 100.0% | 100.0% | 100.0% | 100.0% | 100.0% | 100.0% | | **80.2%** | **Kappa** | |
| **F1** | 82.4% | 81.4% | 77.0% | 83.2% | 88.6% | 88.9% | 83.6% | 78.3% | | **82.3%** | **Weighted-Mean F1** | |

**Table 4.** Impact of label data errors on model accuracy. Totals and metrics in bold.

| Land Cover | Vis. Id. Class (A) | Label Correct (B) | Class Accuracy (B/A) | Model Correct (C) | Class Accuracy (C/A) | Model Correct When Label Correct (D) | Class Accuracy (D/B) |
|---|---|---|---|---|---|---|---|
| Forest | 130 | 93 | 71.5% | 101 | 77.7% | 83 | 89.2% |
| Grassland | 171 | 80 | 46.8% | 144 | 84.2% | 76 | 95.0% |
| Horticulture | 77 | 76 | 98.7% | 52 | 67.5% | 51 | 67.1% |
| Crop | 115 | 92 | 80.0% | 109 | 94.8% | 90 | 97.8% |
| Plantation | 97 | 92 | 94.8% | 82 | 84.5% | 78 | 84.8% |
| Bare | 87 | 84 | 96.6% | 76 | 87.4% | 73 | 86.9% |
| Water | 59 | 57 | 96.6% | 46 | 78.0% | 44 | 77.2% |
| Built-up | 64 | 58 | 90.6% | 54 | 84.4% | 51 | 87.9% |
| **Total** | **800** | **632** | | **664** | | **546** | |
| **Weighted-Mean Class Accuracy** | | | **67.1%** | | **85.4%** | | **93.4%** |
| **Overall Accuracy** | | **(B/A):** | **79.0%** | **(C/A):** | **83.0%** | **(D/B):** | **86.4%** |

## 4. Discussion

A comparison of label data, U-Net CNN and RF model predictions for the test tile +14, −40 shows that at this scale, the U-Net model results more closely matched the label data than the RF results (Figure 6a).

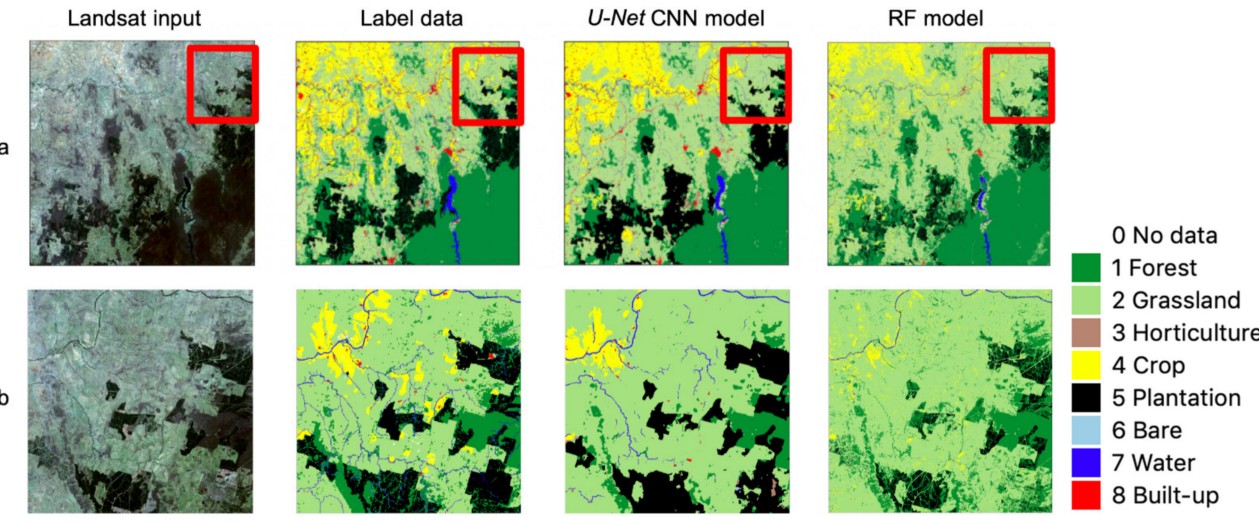

**Figure 6.** (**a**) Landsat input (true colour RGB), label data and model results for tile +14, −40 (~100 km × 100 km). (**b**) An area of tile +14, −40 surrounding the locality of Adjungbilly (box in (**a**)).

Some more fundamental differences between the CNN and RF results become obvious when observed in greater detail. In general, the U-Net model prediction better agreed with the label data for broad classes such as plantation, grassland and crop than the RF model output (Figure 6b). Comparison with the Landsat RGB image provides evidence that U-Net produces better results. Watercourses were reasonably well delineated by the U-Net model and slightly less well by the RF model. On the other hand, the RF model appeared superior in detecting small details, such as roads within plantations. These were generally not found in the U-Net output, which generalises classes to a broader spatial scale. Being a pixel-based method that does not consider spatial context, the RF output featured more 'salt-and-pepper' signal as well as noise, but on the other hand, also tended to better

delineate linear boundaries between classes, such as plantation boundaries. Built-up areas (mainly farm dwellings in this area) were reasonably well detected by the U-Net model, but less so by the RF model. For the cleared and re-planted plantation coupes (south-east of Figure 6b), the U-Net model generally correctly classified this area, with some small areas of misclassification (e.g., horticulture, bare), whereas this area of new plantation was generally undetected by the RF model.

Results for another region (Figure 7a) suggest that the CNN model in fact provided better definition of native forest and plantation than the label data, although plantation boundaries showed some rounding of edges and blunted corners, when compared with the straight edges in the Landsat 8 image. The predicted forest in the central north of the image (red rectangle, Figure 7a) did not appear in the label data, but was seen in the Landsat RGB. Those in the east (box, Figure 7a) were designated as production from native forests in the label data, but inspection of the Landsat imagery showed that these were non-native softwood plantations, correctly detected by the CNN model. These results show the potential of CNNs for improving existing land cover and land use maps.

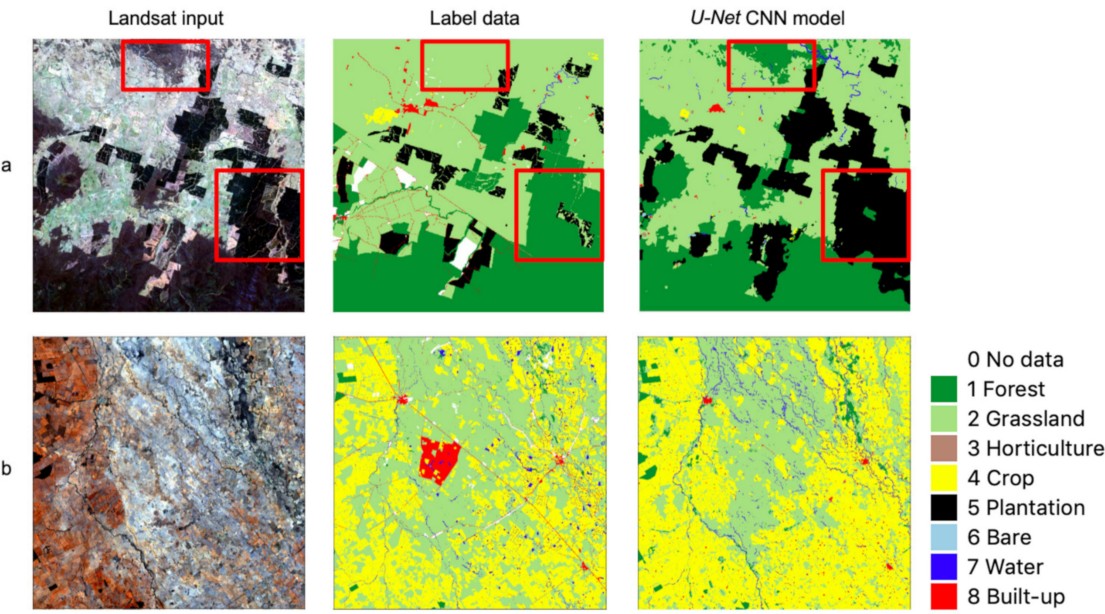

**Figure 7.** (**a**) Landsat input (true colour RGB), label data and U-Net CNN model for area (~26.5 km × 26.5 km) centred at 37.10°S, 149.01°E. (**b**) Results for tile +14, −36 (~100 km × 100 km).

Furthermore, areas that could not be translated unambiguously into label data (white in Figure 7a) could be accurately resolved by the CNN model to a mixture of mainly forest, plantation and grassland.

Results for a third region (Figure 7b) also showed definition of forested areas and water by the CNN model that was generally superior to that in the label data. The large red area in the label data represents a land use class 'Residential and farm infrastructure' that was translated to built-up area. This area was largely mapped to grassland by the U-Net CNN model, which judging by the Landsat imagery was a correct interpretation, even though there were some sparsely placed farm buildings. The CNN model generally slightly overpredicted the extent of cropland, when compared with the label data. This phenomenon was observed in many other areas.

Good correlation was expected between the label data and model output at the scale of the entire study area, as this was where the model was trained (Figure 8). For the whole study area, the CNN model underpredicted the extent of Forest (26.2% vs. 28.2%) and Grassland (31.3% vs. 32.8%), and overpredicted the extent of Crop (33.5% vs. 29.7%), Horticulture (0.6% vs. 0.4%), Plantation (1.6% vs. 1.4%), Bare (0.4% vs. 0.1%) and Water

(4.7% vs. 1.9%) (Supplementary Tables S2 and S6). The extent of the Built-up class was nearly identical between the CNN model and label data (1.9%).

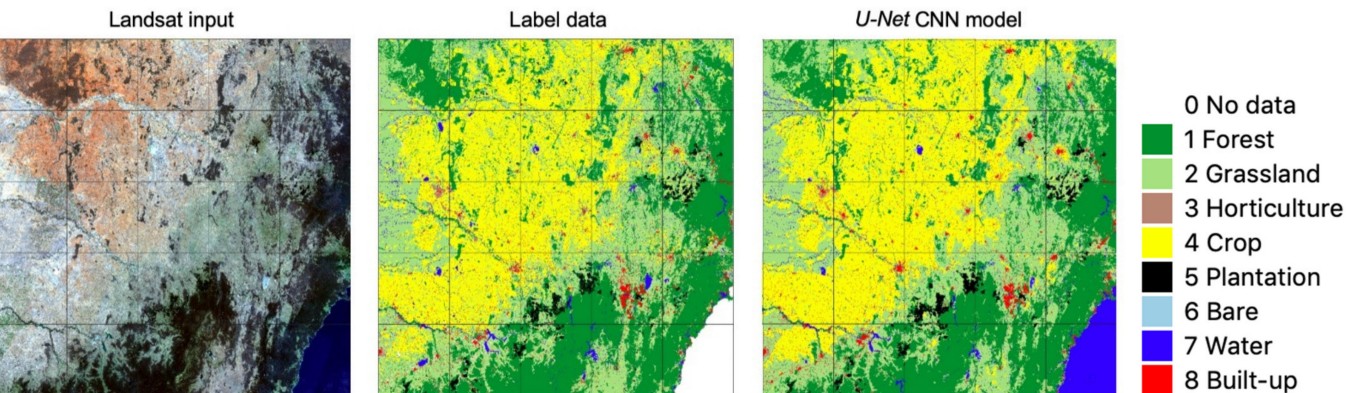

**Figure 8.** U-Net CNN model compared with label data and Landsat input (true colour RGB) for the study area (~500 km × 500 km).

### 4.1. Some Observations on U-Net CNN Model Output

The most accurate CNN model generally represented land cover classes reasonably well at coarse resolution but lacked some details that were more visible in the RF model outputs. For example, linear class boundaries were sometimes poorly defined by the CNN model. As might be expected, spectrally distinct classes (e.g., Forest vs. Grassland, or Plantation vs. Grassland) were more easily delineated than less spectrally distinct classes.

In one example (Figure 9), industrial sites were assigned Built-up and Bare classes in the label data, but were mapped to the Bare class by the CNN model. A quarry at the centre of the image was correctly mapped as Bare. These features were also not easily visually distinguished in the Landsat geomedian data. The higher spatial resolution of the CNN model output (based on 25 m Landsat pixels) was also evident when compared with the 50 m label data.

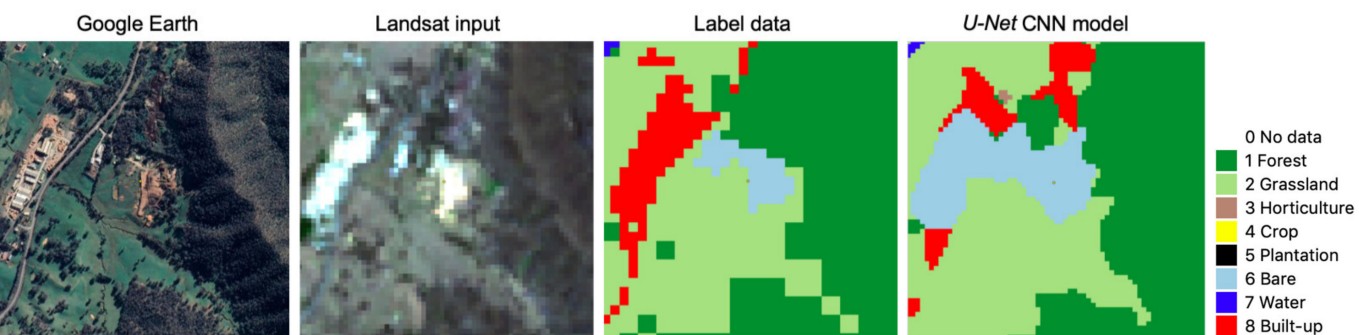

**Figure 9.** Google Earth, Landsat input (true colour RGB), label data and U-Net CNN model for area (~1.6 km × 1.6 km) centred at 35.33°S, 148.18°E.

Some overprediction of the Crop class and lack of definition of the Water class in the model results was also observed (e.g., Figure 10a). In general, the CNN model overpredicted the Crop class. The geomedian reflectances for the whole year did not capture seasonal phenological changes associated with cropping that may be distinct from other land cover types. In the study area, crops are generally harvested in spring or summer. A further experiment might be to divide the annual geomedian data into four seasonal geomedians to test if these could produce a more accurate delineation of cropland.

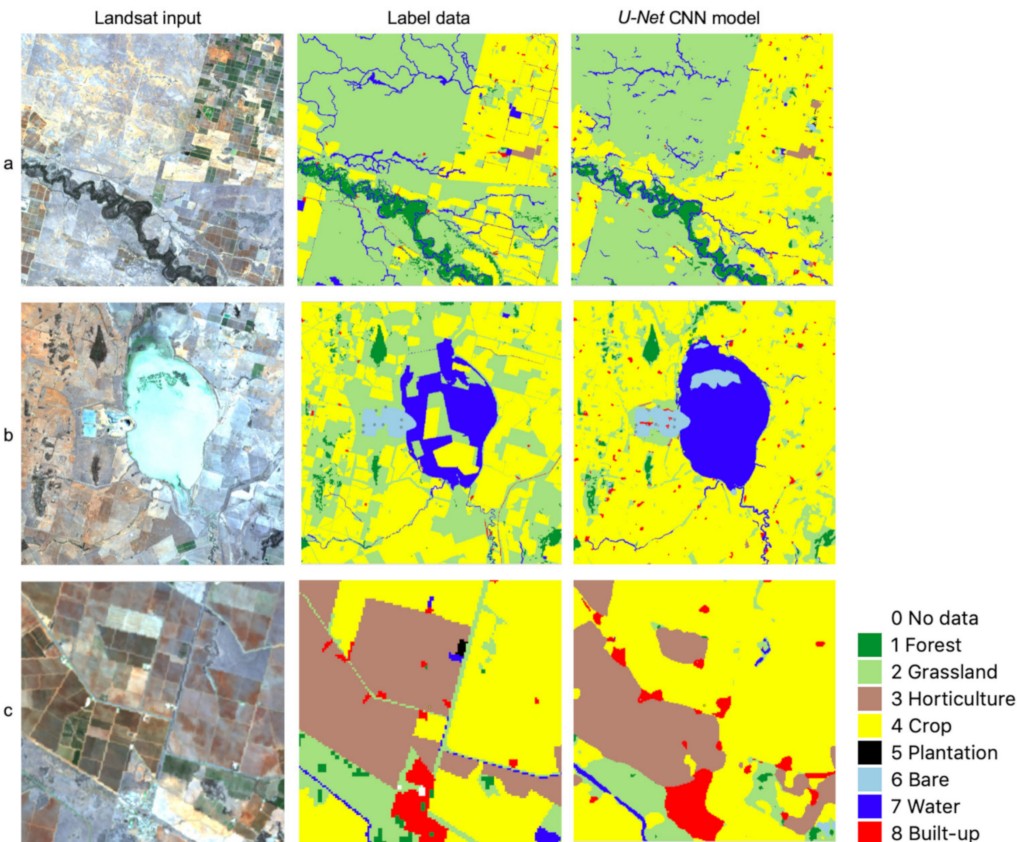

**Figure 10.** (**a**) Landsat input (true colour RGB), label data and U-Net CNN model for area (~28.3 km × 28.3 km) centred at 34.43°S, 145.69°E. (**b**) Results for area (~25.6 km × 25.6 km) around Lake Cowal centred at 33.63°S, 147.45°E. (**c**) Results for area (~6 km × 6 km) centred at 34.49°S, 146.18°E.

The label data included considerable areas of (ephemeral) rivers, creeks, farm dams and lakes in the label data, while no water was evident in the Landsat imagery of the study area for 2018, a much drier than average year. This issue caused errors in the results from the CNN model, with only 78% producer's accuracy for this class (Table 3). For example, in the case of Lake Cowal (Figure 10b), the CNN model assigned the driest areas of the lake to the Bare class and the wetter areas to the Water class. Immediately west of the lake is an open cut gold mine, which was assigned to the Bare class in the label data and also by the CNN model.

The CNN model performed least well for the Horticulture class with 67.5% producer's accuracy (Table 3); considerably less than the next lowest accuracies (Forest, 77.7%; and Water, 78.0%). This was somewhat surprising given the relative ease with which the textured pattern of this land cover type can be discriminated visually, even in Landsat imagery. It may partly reflect errors in the label data: only 76 of the 100 sample points for this class were labelled correctly, while there were many examples of over-extensive or erroneously positioned horticulture areas in the label data (Figure 10c).

## 4.2. Spatial Transferability

The CNN model was applied northeast of the study area (Figure 1) to test how well the trained model transferred to a different area. This 500 km × 500 km area of 25 tiles contained a similar range of land cover types to the study area, ranging from coastal forests in the east through to cropping areas and increasingly sparse grassland in the west (Figure 11). The CNN model appeared to identify forested areas well, with extents closely matching those observed in Landsat 8 and higher resolution imagery. The Water and Built-up classes also appeared broadly correct. The main deficiency was over-representation of the Crop class, especially in the northwest of the area, and some over-representation of the

Bare class in cropping areas. Such over-representation of the Crop and Bare classes did not seem to occur to the same extent further to the east in higher rainfall areas. A possible cause is that grazing land in the more arid western regions can have very sparse vegetation cover, giving it the signature of fallow cropping land or the bare class.

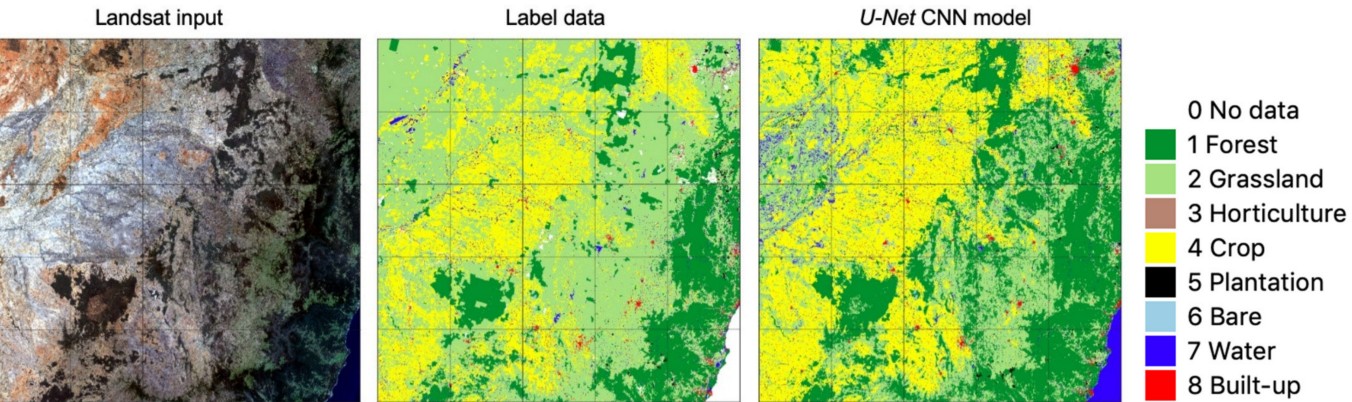

**Figure 11.** Landsat input (true colour RGB), label data and U-Net CNN model prediction for area (~500 km × 500 km) northeast of study area.

## 5. Conclusions

We explored the application of CNNs to undertake broad-scale land cover classification using annual Landsat 8 geomedian surface reflectances [37] for a study area in south-eastern Australia. We compared the results to those of a Random Forests algorithm, representing a machine learning approach that may be considered mainstream. In doing so, we also tested the validity of using proxy data (i.e., land use translated to land cover) to train machine learning models.

With regard to our first objective, we found that CNNs have advantages for automated land cover classification when compared with Random Forests. The most accurate U-Net CNN model showed particular proficiency in the detection of forests, grasslands and crops, which together made up over 90% of the study area. The U-Net CNN model produced superior results to the Random Forest classifier, including the ability to detect contextual information to generate a more consistent spatial distribution of map units at a more aggregated scale, and less pixelated maps.

With regard to the second objective, our results show that a well-performing machine learning classifier can be trained on imperfect label data. Verification of the label data and model results for 800 visually assessed locations suggest that errors in the label data are sometimes compensated for by the trained model. Our work suggests that use of proxy land cover datasets may be an effective approach to generate label data for large-scale remote sensing training datasets for the application of deep learning methods—such datasets are not easy to acquire or in widespread use at present.

Some deficiencies of the U-Net CNN model for segmentation were identified, including lack of detail in segmentation results, lack of definition of linear land cover boundaries and overprediction of the crop class. Future work could attempt to address these issues by testing alternative CNN models for segmentation, such as SegNet [64], feature pyramid networks (FPN) [65] or pyramid scene parsing network (PSPNet) [66]. Alternatively, testing U-Net with larger tiles (e.g., 256 × 256 or 512 × 512 rather than 128 × 128 pixels) to increase the spatial context that can be extracted [67], or different numbers of encoder/decoder levels may yield improved results and better definition of finer spatial details and less common classes.

Land cover maps generated using deep learning CNNs have the potential to be used for quality control and improvement of existing land cover products and for broad-scale simple land cover mapping through time, based on global open access data such as Landsat or Sentinel-2. Future work is envisaged to generate and analyse the effectiveness of a

time series of broad scale annual land cover maps in Australia, and their comparison with existing reference land cover products.

**Supplementary Materials:** The following supporting information can be downloaded at: https://www.mdpi.com/article/10.3390/rs14143396/s1, Figure S1: Autoencoder model summary. Input images are 128 × 128 pixels. Output is a pixel-based land cover classification of nine classes (eight land cover classes plus no data); Figure S2: Accuracy of U-Net CNN model using different ResNet encoders with ImageNet weights; Table S1: Class correspondence of ALUMv8 land use classes and simplified land cover classes; Table S2: Label data statistics—no. pixels of each land cover class for tile +14, −40 and study area; Table S3: Model results—most accurate U-Net CNN and Random Forests model predictions in bold; Table S4: U-Net CNN model (bm_r1_urn50_9b_e34.h5) vs. label data confusion matrices for test tile; Table S5: Random Forests modelling results for six band Landsat geomedian input; Table S6: U-Net CNN model statistics—no. pixels of land cover classes for tile +14, −40 and study area.

**Author Contributions:** Conceptualisation, investigation, methodology, software, writing—original draft, writing—review and editing, T.B.; supervision, visualisation, writing—review and editing, A.V.D.; supervision, writing—review and editing, P.R.L.; supervision, writing—review and editing, R.T. All authors have read and agreed to the published version of the manuscript.

**Funding:** This research was supported by an Australian Government Research Training Program (RTP) Scholarship.

**Acknowledgments:** We thank Rakhesh Devadas (ABARES) for assistance with interpretation of the Catchment Scale Land Use of Australia and Forests of Australia datasets; Bex Dunn and David Gavin (Geoscience Australia) for assistance with Landsat 8 geomedian data and Vasileios Syrris (JRC) for sharing some example code on the application of CNNs to satellite data.

**Conflicts of Interest:** The authors declare that they have no known competing financial interests or personal relationships that could have appeared to influence the work reported in this paper.

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
