# Peer review of "Comparing CNNs and Random Forests for Landsat Image Segmentation Trained on a Large Proxy Land Cover Dataset"

_remotesensing, doi:10.3390/rs14143396_

Round 1

Reviewer 1 Report

Authors implemented and tested UNet model for land cover classification using Landsat 8 bands and proxy training set. The results of UNet is compared with random forests classification. This manuscript is very well written and it checks all the boxes with respect to clear introduction with detailed literature review, analysis, dissemination of results and discussion.

My main concerns with this manuscript are:

Why UNet was chosen among many other deep learning architectures?

Why authors are comparing RF model as bench mark?

Why only comparing two models? Are there other LCC products did authors try to compare? It will be interesting to authors to know that.

A detailed study of land cover classification with different deep learning models would have been a novel work.

Though this is a good work, I cannot recommend this work to be published in MDPI remote sensing in its current form. I suggest authors to add more deep learning frameworks along with UNet and RF.

Reviewer 2 Report

  • Why UNet was chosen among many other deep learning models?
  • Why authors are comparing UNet to Random Forests (RF)? Why not to a reference land cover classified product that has better accuracy.
  • A comparison of multiple deep learning architectures to standard land cover product will be very interesting. 

Reviewer 3 Report

Peer Review

Line 108

Geomedian data as input?? This needs a rather extensive literature backup to proof that this is a working method and has comparable results with respect to the classic “tassled cap” analysis of Landsat data as well as it’s unique panchromatic BAND

The profile of especially Infrared/Red bands during growth season are often crucial to vegetation mapping

Figure 3

128x128  is window size ??? and this does not conflict with OBIA ??? where you have a minimal object size but various shapes ???  ( see also line 227)

Line 207

RGB are highly correlated, N and S1 S2 less  This would suggest to try an indicator as input. Most of the time NDVI as a start

These combinations chosen , the authors might elaborated on the reasons for their choice

Line 210 seems useless or superfluous to split B and G as already correlated to R

Also 252

Pixel accuracy versus accuracy of Image Objects ?? any extra details ??

254  Build-up is often very easy using a Haralick textural image especially from the Landsat Panchromatic band completely missing here

This should be an important argument Why leaving panchromatic band out ????

Modern Remote Sensing works with large stacks of time series of imagery

Mostly on raw data ( geometric and radiometric calibrated)  and sometimes on indices as well as texture

The chosen inputs are derived datasets making it difficult to see how this improves time series analysis as “state of the art” especially with such large free datasets as from Sentinel 2 are available

A very good extension of your study would be to show Landsat next to Sentinel 2 over the same test area

But better reconsider seriously to integrate Panchromatic data and re-discuss geomedian as an input

As Panchromatic ( time series) offer a better base , also for segmentation

I can imagine that the self chosen limitations on your input data makes this study less interesting for state of the art Remote Sensing colleges

Reviewer 4 Report

Review for the ‘Comparing CNNs and Random Forests for Landsat image segmentation trained on a large proxy land cover dataset’

This work is interesting and the proposed method is well presented. I have the following concerns:

1)      In the introduction, the motivations and main contributions should be emphasized clearly. In addition, the application of the CNN and RF on various types of remote sensing data could be discussed, such as the Hyperspectral images and SAR images, see 0.1109/JSTARS.2021.3069013, 10.1109/TGRS.2021.3079438, and Gong M, Zhan T, Zhang P, et al. Superpixel-based difference representation learning for change detection in multispectral remote sensing images[J]. IEEE Transactions on Geoscience and Remote sensing, 2017, 55(5): 2658-2673.

2)      The information of the dataset can be represented using a table, which is more clear.

3)      Why does the method use the U-Net without considering other deep image segmentation models, such as the FCN?

4)      It would be better to discuss the efficiency of U-Net and RF. In my opinion, the RF is very efficient.

Author Response

Note documents for reviewers 2, 3 and 4 are allocated to the wrong reviewers due to a processing error. See covering letter for details.

Round 2

Reviewer 1 Report

Author's response is sound and reasonable, so I can recommend the work to be published.

Reviewer 2 Report

Authors did not make many changes to the manuscript. However I am happy with the response provided by the authors to my comments and questions.

Reviewer 4 Report

I have no comments at this time.